# BFNB Enhances Hair Growth in C57BL/6 Mice through the Induction of EGF and FGF7 Factors and the PI3K-AKT-β-Catenin Pathway

**DOI:** 10.3390/ijms241512110

**Published:** 2023-07-28

**Authors:** Salvador Pérez-Mora, Juan Ocampo-López, María del Consuelo Gómez-García, David Guillermo Pérez-Ishiwara

**Affiliations:** 1Laboratorio de Biomedicina Molecular 1, ENMyH, Instituto Politécnico Nacional, Mexico City 07320, Mexico; sperezm1510@alumno.ipn (S.P.-M.); cgomezg@ipn.mx (M.d.C.G.-G.); 2Laboratorio de Histología e Histopatología (Área Académica de MVZ, ICAp), Universidad Autónoma del Estado de Hidalgo, Tulancingo de Bravo 43600, Mexico; jocampo@uaeh.edu.mx

**Keywords:** BFNB, hair growth, folliculogenesis, growth factors, PI3K-AKT-β-catenin, cell cycle

## Abstract

The objective of this study was to investigate the potential effects of a formulation derived from the bioactive fraction of nanostructured *Bacopa procumbens* (BFNB) on the promotion of hair growth in C57BL/6 mice. The characterization of the follicular phases and histomorphological analysis showed that the topical application of the formulation for 15 days significantly increased pigmentation and hair growth on the dorsum and head of the mice. Additionally, an acceleration of the follicular cycle phases was observed, along with an increase in the number of follicles, hair length, and diameter, compared to mice treated with minoxidil. In silico analysis and molecular characterization demonstrated that BFNB enhances the expression of epidermal growth factor (EGF) and fibroblast growth factor 7 (FGF7), activating the PI3K-AKT-β-catenin signaling pathway, as well as the expression of PCNA, KI-67, Cyclin D1, and Cyclin E, regulating the cell cycle and cell proliferation, crucial events for hair regeneration. Our results strongly suggest the utility of BFNB as a therapeutic alternative to stimulate hair growth and promote hair health.

## 1. Introduction

Hair growth is a continuous cyclic process consisting of four phases: growth (anagen), regression (catagen), rest (telogen), and shedding (exogen) [1,2]. Abnormal hair loss, known as alopecia, is caused by an alteration in the phases of the hair follicle, which can be triggered by hormonal imbalance, stress, nutritional deficiencies, UV radiation, and aging, among other factors. These factors affect scalp health and can contribute to hair loss [3,4,5].

According to data from the American Academy of Dermatology, androgenetic alopecia is the most prevalent form of hair loss, affecting approximately 50% of men and 30% of women; alopecia areata is the second most common type, affecting about 2% of the global population [6,7,8,9].

Minoxidil and finasteride are the only FDA-approved drug treatments for alopecia; however, they are associated with side effects that limit their long-term use. Minoxidil can cause irritative dermatitis and transient facial hypertrichosis, while finasteride may lead to sexual dysfunction, infertility, and other adverse effects [10]. Consequently, the search for novel alternatives that can promote hair growth remains a priority. Plants represent a valuable source of diverse compounds for treating various pathologies, and they have recently garnered significant interest in the cosmetic industry for their potential application in skin and hair care [11,12].

It has been demonstrated that plant extracts can stimulate hair growth by activating the cell cycle and increasing the expression and activation of various growth factors, such as epidermal growth factor (EGF), fibroblast growth factor 7 and 2 (FGF 7-2), vascular endothelial growth factor (VEGF), keratinocyte growth factor (KGF), insulin-like growth factor 1 (IGF), and hepatocyte growth factor (HGF) [13,14,15,16,17,18,19,20,21,22,23,24,25]. 

Recently, we found that an extract of *Bacopa procumbens* (*B. procumbens*) promotes skin regeneration and accelerates the wound-healing process [26]. 

In the present study, we evaluated the promoter effect of a formulation derived from the bioactive fraction of nanostructured *B. procumbens* (BFNB) in C57BL/6 mice. The results obtained suggest that topical application of the formulation accelerates the follicular phases, enhances hair growth, increases the number of hair follicles, and thickens the hair. Molecular and interactomics analyses suggest that BFNB upregulates the expression of EGF and FGF7, activates the PI3K-AKT-β-catenin signaling pathway, and enhances the expression of PCNA, KI-67, Cyclin D1, and Cyclin E, thereby stimulating hair growth. 

## 2. Results

### 2.1. Effect of a Nanostructured Formulation of B. procumbens on Hair Growth Regulation and Follicular Phase Dynamics in the Dorsa of C57BL/6 Mice

The results shown in Figure 1a illustrate the temporal progression of hair growth for 21 days following treatment application. At day 10, an evident increase in skin pigmentation was observed on the dorsal regions of mice treated with the BFNB formulation, exhibiting a 10.2-fold higher intensity compared to the control group and a 5.1-fold higher intensity than mice treated with minoxidil (Figure 1b). We observed a remarkable and statistically significant improvement in hair growth on days 15, 18, and 21 in animals treated with the bioactive compound, surpassing the hair growth observed in animals treated solely with the vehicle or minoxidil.

The average initial hair length of the mice before depilation was 7.6 mm, a length achieved by the bioactive-treated mice on day 21, while the control group and minoxidil-treated mice reached lengths of only 3.1 mm and 6.4 mm, respectively (Figure 1c).

Analysis of the hair follicles revealed that the application of the bioactive compound accelerated the transition process between follicular phases. The anagen phase was identified on days 15 and 18, the catagen phase persisted from day 21 to 24, and the telogen phase extended from day 27 to 30 in animals treated with BFNB or minoxidil. By contrast, animals treated with the vehicle displayed the anagen phase from day 18 to 21, the catagen phase from day 24 to 27, and the telogen phase on day 30 (Figure 2a).

Interestingly, follicular size in terms of length (Figure 2b) and diameter (Figure 2c) was significantly larger in mice treated with our formulation compared to those treated with minoxidil or the vehicle on day 15. Similarly, follicles in mice treated with the BFNB formulation or minoxidil showed signs of resorption, indicating the transition to the catagen phase, which was evident on day 18 by the rupture of the inner root sheath cells. By contrast, mice in the control group exhibited a delay in the transition of follicular phases, as the disruption of inner root sheath cells was not observed until day 24. These findings suggest that the vehicle used as a control does not have a stimulating effect on hair growth compared to the BFNB formulation.

### 2.2. Effect of BFNB Formulation on the Modulation of Hair Growth and Follicular Phase Dynamics on the Heads of C57BL/6 Mice

Similar to the response observed in the dorsal area, the pigmentation and hair growth on the heads of C57BL/6 mice were stimulated by the formulation (Figure 3a). Pigmentation started to become visible from day 10 in mice treated with the bioactive compound, whereas animals treated with vehicle or minoxidil lacked visible pigmentation in the area. On day 15, significantly higher pigmentation was observed in the group treated with the BFNB formulation, being 8.5-fold higher than in the vehicle-treated mice and 10-fold higher greater than in the minoxidil-treated mice (Figure 3b). As expected, a significant increase in hair growth induced by the bioactive compound was also observed from day 15 compared to the other groups. Hair in mice treated with the formulation achieved a length of 7.3 mm on day 21, comparable to the initial length (before depilation, 7.5 mm), while the other two experimental groups reached lengths of 2.6 mm and 5.2 mm, respectively (Figure 3c).

In a similar way to the dorsal area, BFNB and minoxidil were found to stimulate the hair follicle cycle on the head. With both treatments, the anagen phase was observed on day 15, followed by the catagen phase from day 18 to 21, and the telogen phase from day 24 to 30. By contrast, the vehicle-treated group exhibited anagen follicles until day 18, catagen from day 21 to 30, and telogen after this time (Figure 4a). Interestingly, the follicles in mice treated with the BFNB formulation were longer and thicker on days 15 and 18 compared to the controls, and from day 21, their resorption became noticeable, indicating an acceleration of the follicular phases induced by the bioactive compound, similar to the effect observed in the dorsal area (Figure 4b,c).

### 2.3. Histological Analysis of the BFNB Effect on the Dorsa of C57BL/6 Mice

Histological analysis of the animals after 10 and 15 days of treatment with BFNB showed a significant increase in the number of hair follicles compared to the control group or minoxidil-treated animals. The length and diameter of the hair follicles after 10 and 15 days of treatment with BFNB suggest that the hair follicles not only increased in number but were also significantly longer and wider than those in the other experimental groups (Figure 5a,b). In addition, we also observed that the epidermis of mice treated with the bioactive compound showed increased thickness compared to those receiving minoxidil or vehicle after 15 days of application. By contrast, a significant reduction in dermal thickness was observed at day 15 in mice treated with BFNB. Meanwhile, the hypodermis was thicker in mice that received BFNB or minoxidil on days 10 and 15 compared with the control group (Figure 5b).

### 2.4. In Silico Prediction of Signaling Pathways Related to the Modulation of Hair Growth by EGF and FGF7

Using in silico prediction, we analyzed the molecular interaction networks of EGF and FGF7 growth factors to identify the pathways involved in hair growth (Figure 6). EGF interacted with 50 proteins with a probability of interaction greater than 90%, including the EGFR receptor, growth factors IGF1 and FGF2, and several proteins involved in cell proliferation, such as PI3K, CTNNB1 (β-catenin), mTOR, and RAS. FGF7 interacted with 35 proteins with a probability of interaction greater than 70%, including the FGFR2, FGFR3, and FGFR4 receptors, as well as proteins related to vascularization, such as VEGF and TED, and some proteins involved in the proliferation pathway, like RAS and MAPK. These results suggest that EGF and FGF7 can modulate fundamental molecular pathways in hair growth.

### 2.5. Immunodetection of Hair-Growth-Inducing Proteins

Taking into consideration the interatomic analyses and the relevance of specific proteins known to be involved in hair growth induction, we proceeded to analyze their expression in dorsal and head skin samples through Western blot and immunohistochemistry.

In the Western blot assays, a significant increase in the expression of EGF and FGF7 factors was observed in animals treated with BFNB compared to the control group, both in the dorsal and head regions. Furthermore, it was found that the BFNB formulation stimulates the PI3K-AKT pathway, as demonstrated by the increased expression of activated (phosphorylated) proteins PI3K-pTyr467/pTyr19 and AKT-pS473. When evaluating crucial proteins in the cell cycle, it was also observed that BFNB promoted the expression of PCNA, Ki-67, Cyclin D1, Cyclin E, and β-catenin compared to the control group in both body regions studied. Similarly, mice treated with minoxidil showed a marked increase in the expression of the EGF, FGF7, PI3K-pTyr467/pTyr19, AKT-pS473, PCNA, Ki-67, Cyclin D1, Cyclin E, and β-catenin proteins in the dorsal region compared to the vehicle-treated group. In addition, in the head, we found a significant increase in the PI3K-pTyr467/pTyr19, AKT-pS473, and Cyclin D1 proteins. Remarkably, the expression of these proteins was considerably higher in mice treated with BFNB compared to those treated with minoxidil. (Figure 7 and Table 1).

To correlate the expression, localization, and distribution of the EGF, FGF7, PCNA, and β-catenin proteins, immunohistochemistry assays were performed on longitudinal and transverse histological sections of the dorsal skin of mice 15 days after the treatments. In both histological sections, a significant increase in the expression of the evaluated proteins was again observed in mice treated with the BFNB formulation compared to the other experimental groups. It is noteworthy that these proteins showed a predominant distribution in the epidermis, dermis, and adipocytes of the hypodermis. EGF, FGF7, PCNA, and β-catenin were also notably expressed in the cells composing the hair follicle. By contrast, in the control group, the proteins of interest were predominantly immunodetected in the dermis, while in the mice treated with minoxidil, mainly in the dermis and to a lesser extent in the adipocytes of the hypodermis (Figure 8 and Table 2). The results obtained through these molecular techniques are consistent and support the potential of our formulation as an effective modulator of the expression of essential proteins in the stimulation of hair growth.

## 3. Discussion

In this work, we present histomorphological and molecular evidence suggesting that a nanostructured bioactive compound from *B. procumbens* (BFNB) has an enhancing effect on hair growth. Additionally, we observed that BFNB induced early skin pigmentation in C57BL6 mice, surpassing the effects observed in the groups of mice that received either the vehicle or minoxidil. The presence of pigmentation in C57BL6 mice has been widely accepted as an early indicator of hair growth, as it signals the presence of an active proliferative process or anagen phase, while the absence of pigmentation indicates that the hair follicle is in the telogen phase [27,27,28]. The results presented here suggest that the BFNB formulation has the potential to accelerate and promote the transition from the telogen phase (depilated mice) to the anagen phase (pigmented mice), which could be an enormously valuable first step in the treatment of alopecia. This fact correlates significantly with the acceleration of the phases of the hair follicle cycle that we observed when analyzing them in the body areas subjected to treatment. Complementarily, the histological analysis showed that BFNB not only accelerates the phases of the follicular cycle but also increases follicle size (length and diameter) after 15 days of application, in addition to increasing the number of hair follicles at days 10 and 15 post-treatment, compared to mice treated with vehicle or minoxidil. This strongly suggests that our formulation stimulates folliculogenesis and strengthens hair follicles in the first instance by thickening them and possibly making them more resistant to shedding induced by external agents.

The proliferative and folliculogenesis-promoting effects have been reported in other plant-derived extracts in the employed murine model, such as *Polygonum multiflorum* extract [29], *Nelumbinis semen* extract [30], and *Panax ginseng* Fructus extract [31]. When comparing the topical effect of the BFNB formulation with these extracts, we observed that BFNB induces pigmentation at earlier times. Furthermore, the hair length in the dorsal area reached 7.24 mm on day 21 in mice treated with our formulation, which is similar to the pre-depilation hair length. By contrast, the hair length with the plant extracts was lower, ranging from approximately 4.5 to 6.5 mm. 

Another recent study conducted by Begum et al. [32] aimed to develop a 1% hair lotion utilizing compounds extracted from *Rosmarinus officinalis* (*R. officinalis*). The results demonstrated that this topical formulation exhibited similar effects to BFNB. Notably, pigmentation was observed in mice treated with the lotion at 8 days. At 14 days, superior hair growth was observed, and by 3 weeks, hair growth was complete. Histological analyses further revealed a greater number and thickness of follicles in mice treated with the lotion at 14 days. However, upon a visual and quantitative comparison of the histology between mice treated with *R. officinalis* and BFNB, our study exhibited a higher number of follicles per histological field. In mice treated with *R. officinalis*, approximately 60 follicles were observed after 15 days of treatment, whereas with BFNB, at 10 days, the number of follicles exceeded 70 per histological field.

The epidermis, dermis, and hypodermis provide valuable insights into cellular activity and perifollicular vascularization during different phases of the hair follicle cycle, offering a better understanding of scalp processes [33,34]. After histological analysis of these skin layers, we observed that BFNB increased the thickness of the epidermis and hypodermis, while the thickness of the dermis decreased after 15 days of application. These findings suggest a high rate of cellular proliferation, leading to the thickening of the epidermis and hypodermis.

This thickening has several advantages, as a thicker epidermis in the scalp provides enhanced protection and resistance against environmental factors and external aggressions, serving as an additional physical barrier to prevent moisture loss and the penetration of irritants or harmful agents, such as UV rays, harsh chemicals, or pathogenic microorganisms [35,36]. In addition, a thicker hypodermis plays a supportive role by providing a reinforced layer of fatty tissue under the dermis and the hair follicles. This provides thermal isolation and additional protection to the follicles. Moreover, it indirectly, positively influences hair growth by regulating hormonal processes and facilitating the supply of necessary nutrients [37,38,39,40,41]. These findings could perfectly correlate with our results, as we found that BFNB induces folliculogenesis after 15 days of treatment, which, in turn, promotes the thickening of the epidermis and hypodermis to provide greater support and protection to the hair follicles.

The hair growth observed in our study may be strongly related to the findings of Martínez-Cuazitl et al. [26] regarding the healing process. Their research highlighted the regulatory effect of the bioactive fraction from *B. procumbens* on cell proliferation, adhesion, and migration, as well as its ability to reduce the area of artificial wounds in 3T3 fibroblasts in vitro. Interestingly, when this bioactive fraction was incorporated into a hydrogel and topically applied to Wistar rats with skin wounds, it accelerated the healing process on days 5 and 7 post-treatment. This acceleration was associated with an increase in the levels of collagen type I and III, as well as the modulation of proteins such as PCNA, ERK1/2, TGF-β1, and p-SMAD 2/3. These proteins play crucial roles in various processes involved in skin regeneration, including cell proliferation, differentiation, cell cycle regulation, migration, survival, and growth control. These results provide strong support for the potential of bioactive compounds derived from *B. procumbens* not only to expedite the healing process but also to stimulate hair regeneration.

It is important to emphasize the capacity of alopecia treatments to promote the transition from the telogen phase to the anagen phase, a critical stage in the hair growth cycle. Several studies have demonstrated that this transition is regulated by several proteins released by hair follicle cells, including growth factors, which work synergistically to stimulate the proliferation of hair follicle cells and facilitate hair growth during the anagen phase [13,14,15,16,17,18,19,20,21,22,23,24,25,34,42,43,44,45,46].

FGF-7, also known as HGF, is a growth factor released by dermal papilla cells, dermal cells, and to a lesser extent, epidermal cells. Previous research has elucidated its role in hair follicle growth, differentiation, and morphogenesis [14,15,16,18,22,25,47,48,49,50]. Another crucial growth factor is EGF, which plays a role in the proliferation, differentiation, and migration of cells within the outer root sheath of the hair follicle [18,24,51,52,53,54]. 

Considering the significance of these factors, we conducted in silico analysis of the interaction networks of these proteins to predict the molecular pathways that could be modulated by BFNB formulation. The results suggest that EGF and FGF-7 may modulate molecular pathways critical for hair growth. Based on the results from STRING, we observed a high probability of interaction between these growth factors and key proteins involved in cell proliferation, angiogenesis, and the cell cycle, such as PI3K, β-catenin, mTOR, RAS, MAPK, IGF1, FGF2, VEGF, and TED.

Interestingly, after 15 days of topical application with the bioactive compound, the EGF and FGF-7 proteins exhibited a significant increase in the studied areas of the body, compared to mice that received the vehicle or minoxidil. These results are consistent with previous studies, such as the one conducted by Lee et al. [55], which demonstrated that the food supplement BeauTop, consisting of six plants, stimulates hair growth and early pigmentation through FGF-7 and EGF factors in the same murine model. In a clinical study carried out by Grothe et al. [56], it was shown that pea shoot extract significantly improves hair density and thickness, with increased expression of FGF-7 attributed to the observed effect. In addition, Zhou et al. [57], demonstrated that autologous platelet-rich plasma enhances hair growth in men with androgenic alopecia, highlighting the increase in EGF and FGF-7, as well as activation of the Wnt/β-catenin pathway. Conversely, when EGF and FGF-7 factors are inhibited, hair follicle phases are disrupted, and hair growth is affected [54,58].

Both EGF and FGF7 have been implicated in the activation of the PI3K-AKT-β-catenin signaling pathway in hair follicle cells, leading to hair growth promotion [18,55,58,59]. Consistent with these findings, our study demonstrated that 15 days of BFNB application increased the expression of the PI3K-pTyr467/pTyr199, AKT-pS473, and β-catenin proteins in both the dorsal and head regions, indicating the activation of the PI3K-AKT-β-catenin pathway. This activation resulted in a significant positive modulation of the cell cycle, as evidenced by increased expression of key regulatory proteins such as PCNA, KI-67, Cyclin D1, and Cyclin E.

In a study by Chen et al. [60], β-catenin was identified as a crucial modulator of genes involved in several aspects of hair follicle biology, including hair root stem cells, hair follicle precursor cells, hair bulb cells, and outer root sheath cells. These genes include those related to the cell cycle (Lef-1, Cyclin D1, Cyclin E, and Cdc25 A), growth factors of the FGF family, angiogenesis regulators (VEGF), as well as transcription factors (Sox2, Nfatc1, c-Myc, C-jun, and PPARδ) responsible for cell pluripotency, differentiation, and proliferation [61,62,63,64,65,66,67,68]. 

Furthermore, immunohistochemistry assays revealed a significant increase in EGF, FGF7, PCNA, and β-catenin protein expression in the dorsal tissue of mice treated with the BFNB formulation after 15 days of application. These proteins were detected mainly in the epidermis, dermis, hypodermis, and the periphery of the hair follicles and adipocytes. By contrast, mice treated with vehicle or minoxidil showed attenuated expression of these proteins, predominantly located in the dermis. It has been proposed that these four proteins may act together to create a favorable microenvironment within the hair follicle, favoring hair growth [14,65,69,70].

A great deal of evidence has demonstrated that fibroblasts and keratinocytes, the predominant cellular constituents of the dermis, possess the capability to synthesize and release a plethora of growth factors, including FGF7 and EGF, among others [71,72,73]. These growth factors are considered to play a pivotal role in initiating the cell cycle and facilitating cell proliferation, thereby triggering early pigmentation, and accelerating the progression of hair follicle phases, ultimately promoting hair growth, as corroborated by the results of our study.

It is interesting to highlight that minoxidil also increased the expression of EGF and FGF7 in the dorsal region of mice after 15 days of treatment, as well as the expression of phosphorylated PI3K and AKT, and β-catenin, compared to the control group. However, these expressions were lower than those found with BFNB, both in the dorsal region and in the head. This could be due to the fact that the mechanism of action of minoxidil is based on other molecular pathways, mainly involving the stimulation of microcirculation and vascularization by inducing the expression of the VEGF factor. Additionally, it has been discovered that minoxidil induces the activation of prostaglandin-endoperoxide synthase, inhibits the effects of androgens, and can act as an epidermal growth factor on matrix cells, delaying their aging [74].

In our previous investigations, we meticulously identified the principal secondary metabolites present within the bioactive fraction derived from *B. procumbens* [26]. Naringenin, equol, paeoniflorin, and apigenin were the most abundant components, constituting 29.2%, 19.3%, 8.2%, and 5.6% of the fraction, respectively. The effects of naringenin were thoroughly assessed in vitro employing human follicle dermal papilla cells (HFDPC) and keratinocyte cells (HaCaT), which are crucial components contributing to hair morphogenesis and growth. It was found that naringenin not only stimulated cell proliferation in both cell types but also induced VEGF expression, a pivotal angiogenic factor, while simultaneously conferring cellular protection against the damaging effects of hydrogen peroxide [75]. Conversely, equol was subjected to clinical investigation in individuals suffering from androgenic alopecia, and its ability to inhibit the enzyme 5 alpha-reductase, an essential mediator of hair loss, was established, thus favoring hair regrowth [76]. Apigenin, another prominent metabolite, exhibited a remarkable ability to promote proliferation in HFDPC and HaCaT cells, in addition to facilitating hair elongation in rat vibrissa follicles [77]. Moreover, a study by Dong et al. [78] revealed that paeoniflorin possesses the potential to stimulate tissue regeneration and induce angiogenesis through the activation of the PI3K-AKT pathway. Furthermore, research in non-follicular cell lines has consistently demonstrated the ability of naringenin [79,80], equol [81], and apigenin [82] to modulate the PI3K-AKT-β-catenin pathway. The comprehensive results obtained in this study suggest that the major metabolites present in BFNB have the potential to exert synergistic effects in promoting hair growth. Experiments currently in progress will undoubtedly provide further information on the distinct contribution of each metabolite to the mechanisms that modulate hair regeneration and overall hair health.

In summary, in Figure 9, we propose a possible mechanism of action for the components of the BFNB formulation in stimulating hair growth, highlighting the importance of modulating the growth factors EGF and FGF7, as well as the PI3K-AKT-β-catenin pathway and its positive effect on hair growth. These results suggest that the effect of BFNB could be considered an excellent alternative for efficiently promoting hair growth, presenting an innovative perspective for the use of nanotechnology in the treatment of hair disorders. Furthermore, the findings obtained in this study may open new possibilities for alopecia therapy and future research in hair regeneration. 

## 4. Materials and Methods

### 4.1. Lipophilic Serum Containing the Nanostructured BFNB Formulation

Gold nanoparticles (AuNPs) were obtained following the method described by Turkevitch in 1951. In brief, 0.5 mL of a 4% HAuCl4 solution was added to 200 mL of deionized water and boiled with constant stirring until the sample reached 97 to 100 °C. Subsequently, 3 mL of a 1% sodium citrate solution was added and incubated for 30 min. After cooling, the AuNPs were centrifuged at 3500 rpm for 40 min, the supernatant was removed, and the nanoparticles were resuspended in 6 mL of deionized water. Next, the AuNPs were mixed with 0.8 mg/mL of the bioactive fraction of *B. procumbens* until a homogeneous mixture was obtained. This AuNPs/*B. procumbens* mixture was then added to a lipophilic dermatological-grade serum composed of glycerin, dicaprylether, dicapryl carbonate, caprylic triglycerides, decyl oleate, carbomer, and iodopropyl butylcarbamate to generate BFNB. 

### 4.2. Animals

Male C57BL/C mice were maintained under strict temperature conditions (22 ± 1 °C), humidity (55 ± 5%), and a photoperiod of 12 h light and 12 h darkness cycles, with free access to water and a standard diet. All experimental procedures were approved by the Bioethics Committee of the ENMH-IPN, in compliance with the technical specifications for the care and use of laboratory animals established in the Official Mexican Standard NOM-062-ZOO-1999 [83].

### 4.3. Hair Growth in C57BL/6 Mice

Three experimental groups of 10 mice each were randomly formed. Group I received treatment with the vehicle (a dermatological serum without the *B. procumbens* bioactive compound). Group II was treated with 2% minoxidil (Thermo Fisher Scientific, Waltham, MA, USA, #J61803.03), while Group III received treatment with a dermatological serum supplemented with 8 µg/mL (micrograms per milliliter) of BFNB. At the start of the experiment (day zero), seven-week-old mice were depilated on both the back and the head. Subsequently, the assigned treatments were topically applied every 24 h for 30 days to both body regions. Photographic monitoring of hair growth was performed on days 0, 10, 15, 18, and 21. Before depilation and photography, the animals were anesthetized following the procedures detailed in Sections 8.1 and 8.1.1 of the Official Mexican Standard NOM-062-ZOO-1999 [83]. The administration was carried out intraperitoneally, using a mixture of ketamine/xylazine at doses of 70 and 10 mg/kg, respectively. To evaluate skin pigmentation, photographs taken on day 10 were subjected to densitometric analysis. These images were processed using ImageJ software version 1.53K, developed by the United States National Institutes of Health (NIH). By analyzing the pixel values obtained, the corresponding relative densitometry of skin pigmentation was calculated. The same software was employed to determine hair length.

### 4.4. Follicular Analysis

To evaluate the phases of hair follicles, we collected hair tufts from the dorsum and head on days 15, 18, 21, 24, 27, and 30. These tufts were placed on glass slides and moistened with distilled water. Using an optical microscope equipped with a DP21 photographic system (Olympus, Tokyo, Japan), we captured photographs of the follicles. The obtained images were then processed with ImageJ software (version 1.54) to accurately measure the length and diameter of the hair follicles.

### 4.5. Histological Study

Skin sections were collected from the dorsal regions of mice from the three experimental groups on days 10 and 15 post-treatment. Samples were fixed in phosphate buffer with 3.7% formaldehyde (pH 7.4) at room temperature. The tissues were processed in MICROM/STP120-1 equipment (Thermo Scientific, Walldorf, Germany) for 16 h and embedded in Paraplast (McCormick) (Medex Supply, Brooklyn, NY, USA) blocks, and then longitudinal and transverse sections of 8 µm thickness were made. The histological slices obtained were stained with hematoxylin and eosin (H & E). Histological evaluation and photographic capture were performed in an optical microscope coupled to a DP21 photographic system (Olympus, Tokyo, Japan), utilizing 10× and 40× objectives.

### 4.6. Interactomics of EGF and FGF7 Growth Factors

The protein–protein interaction network for the growth factors EGF and FGF7 was generated using the STRING server (version 11.5) (https://string-db.org/, acceded on 2 January 2023). The interaction networks were created independently for each protein in the *Mus musculus* species. Predictions were made with an interaction probability threshold ≥ 70% and a network of up to 50 proteins.

### 4.7. Western Blotting

Dorsal and head skin samples were collected 15 days post-treatment. Tissue homogenization was performed utilizing a tissue homogenizer (Ultraturrax-T18 digital-IKA, San Diego, CA, USA) at 9000 rpm for 5 min to achieve cell lysis in RIPA buffer (0.1% SDS, 1% NP-40, 50 mM Tris, pH 7.5, 150 mM NaCl, 1 mM EDTA) supplemented with complete™ protease inhibitor cocktail (MERCK, Darmstadt, Hesse, Germany, # 04693116001). The mixture was centrifuged at 13,000 rpm for 15 min at 4 °C, and the soluble fraction was collected. Protein quantification was performed employing the Bradford method [84]. An amount of 20 µg of protein from each group was subjected to 12% polyacrylamide gel electrophoresis (SDS-PAGE), transferred to polyvinylidene fluoride (PVDF) membranes, and blocked with 5% skim milk for 30 min. The membranes were then incubated with the primary antibody at a dilution of 1:15,000 for 30 min, followed by incubation with the secondary antibody conjugated to horseradish peroxidase (HRP) at a dilution of 1:30,000 at room temperature. The detection was carried out using the solutions from the Millipore kit (Immobilion Western, Chemiluminescent HRP Substrate).

The primary antibodies used were anti-EGF (Santa Cruz Biotechnology, Santa Cruz, CA, USA, #sc-1343), anti-FGF7 (Santa Cruz, #sc-7882), anti-PI3K-pTyr467/pTyr199 (GeneTex, Irvine, CA, USA, #GTX132597), anti-AKT (GeneTex, #GTX121937), anti-AKT-pS473 (GeneTex, #GTX128414), anti-PCNA (Santa Cruz, #sc-53407), anti-Ki-67 (GeneTex, #GTX20833), anti-Cyclin D1 (Santa Cruz, #sc-753), anti-Cyclin E (Invitrogen, Carlsbad, CA, USA, #MA5-14336), and, as endogenous control, anti-GAPDH (Santa Cruz, #sc-32233). The secondary antibodies used were anti-mouse IgG (Jackson ImmunoResearch, West Grove, PA, USA, #115-035-062) or anti-rabbit (Jackson, #111-035-003) coupled to horseradish peroxidase (HRP), depending on the origin of the primary antibody.

### 4.8. Immunohistochemistry

Histological sections of 8 µm thickness from the dorsal regions of mice subjected to the 15-day treatments were used for immunohistochemical assays. The samples were deparaffinized in xylene, followed by hydration in a series of descending concentrations of ethanol (100%, 95%, 90%, 80%, and 70%), according to the protocol used by Martínez-Cuazitl et al. [26], with slight modifications and utilizing the solutions from the Mouse/Rabbit PolyDetector DAB HRP Brown kit from Bio SB (Santa Barbara, CA, USA). The tissues were washed with PBS, and epitope retrieval was performed utilizing an Immune Retriever Citrate solution in a pressure cooker (Oster, FlavorMaster^TM^, Owosso, MI, USA) for 8 min. The tissues were cooled to room temperature for 30 min with gentle shaking, followed by a wash with PBS. Subsequently, endogenous peroxidase activity was blocked by employing Peroxidase Block quenching buffer for 10 min. After another wash with PBS, nonspecific binding sites were blocked with 5% BSA (albumin, bovine serum, fraction V, #1328A) for 30 min. Next, the samples were incubated for 1 h in a humid chamber with primary antibodies, anti-EGF, anti-FGF7, anti-PCNA, or anti-β-catenin, diluted 1:200. After three washes with PBS, the samples were incubated for 30 min in a humid chamber with the secondary antibody (HRP PolyDetector), followed by three more washes with PBS. Each PBS wash (pH 7.4) was carried out for 5 min. For visualization, the diaminobenzidine (DAB) peroxidase substrate was used. After rinsing the samples with PBS, they were counterstained with Mayer’s hematoxylin (Sigma-Aldrich, St. Louis, MO, USA, #51275) to obtain a blue coloration of the cell nuclei. Finally, the samples were dehydrated in a series of ascending concentrations of ethanol (70%, 80%, 90%, 95%, and 100%), cleared in xylene, and mounted with GVA-mount reagent (Zymed, San Francisco, CA, USA). Images were captured employing a microscope (Nikon, Tokyo, Japan) with 10× and 40× objectives. Quantification of relative protein expression was performed by pixel analysis using ImageJ software (version 1.54).

### 4.9. Statistical Analysis

The graphs were created using GraphPad Prism software version 8.0.1. Statistical analysis was conducted utilizing one-way and two-way analysis of variance (ANOVA) with Tukey’s multiple comparison test. Significance levels were determined employing the APA format (adjusted *p*-value analysis) as follows: * *p* ≤ 0.033, ** *p* ≤ 0.002, *** *p* ≤ 0.001, and # *p* ≤ 0.033. The data are presented as mean ± standard deviation from a minimum of three independent experiments.

## 5. Conclusions

BFNB demonstrated an enhancing effect on hair growth by accelerating follicular phases, increasing the number of hair follicles, and thickening the hair in the heads and dorsa of C57BL/6 mice. These effects are related to the ability of the nanostructured bioactive compound from *B. procumbens* to modulate the expression of growth factors such as EGF and FGF7, as well as the PI3K-AKT-β-catenin pathway and crucial proteins in the cell cycle including PCNA, Ki-67, cyclin D1, and cyclin E. Our study strongly suggests that the BFNB formulation could be a promising therapeutic alternative to stimulate hair growth in humans.

## Figures and Tables

**Figure 1 ijms-24-12110-f001:**
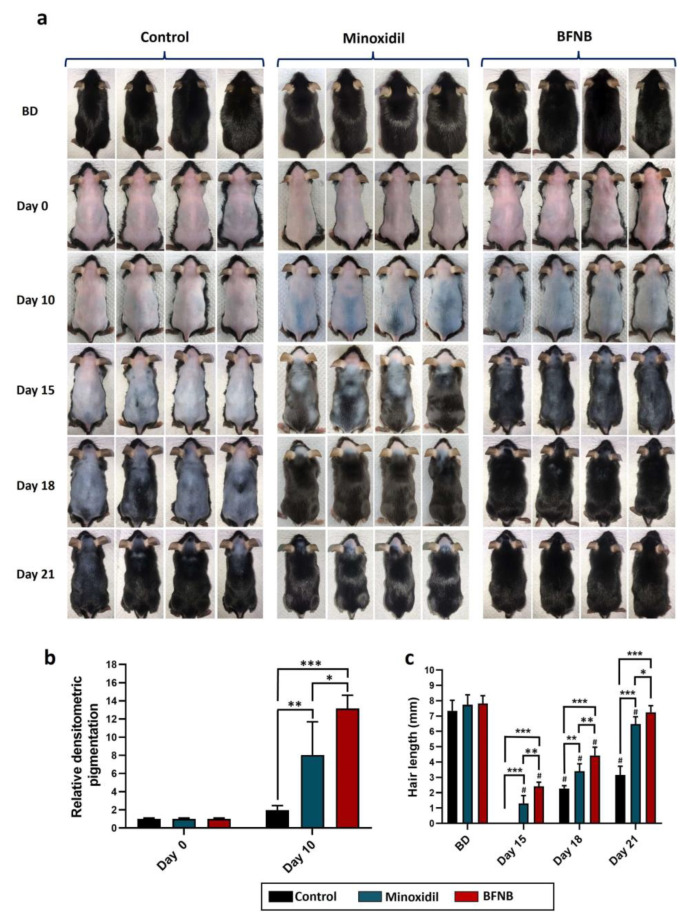
Promoter effect of BFNB on dorsal hair growth in C57BL/6 mice. Evolution of hair growth during the 21 days of treatment (**a**). Densitometric analysis of dorsal pigmentation was conducted on days 0 and 10 post-treatment (**b**). Measurements of hair length were performed at 15, 18, and 21 days after treatment (**c**). The initial hair length before treatment is denoted as BD. Statistical analysis was carried out on the pixel-wise pigmentation data obtained from the skin images of 10 mice per experimental group. To determine hair length, five hair tufts were collected from each of the 10 mice in the different experimental groups. In both cases, a one-way ANOVA was applied, followed by Tukey’s multiple comparison test. The comparison between experimental groups is indicated by (*) and comparisons between groups on days 15, 18, and 21 with BD are denoted by (#). The significance levels are represented as follows: * *p* ≤ 0.033, ** *p* ≤ 0.002, *** *p* ≤ 0.001, and # *p* ≤ 0.033.

**Figure 2 ijms-24-12110-f002:**
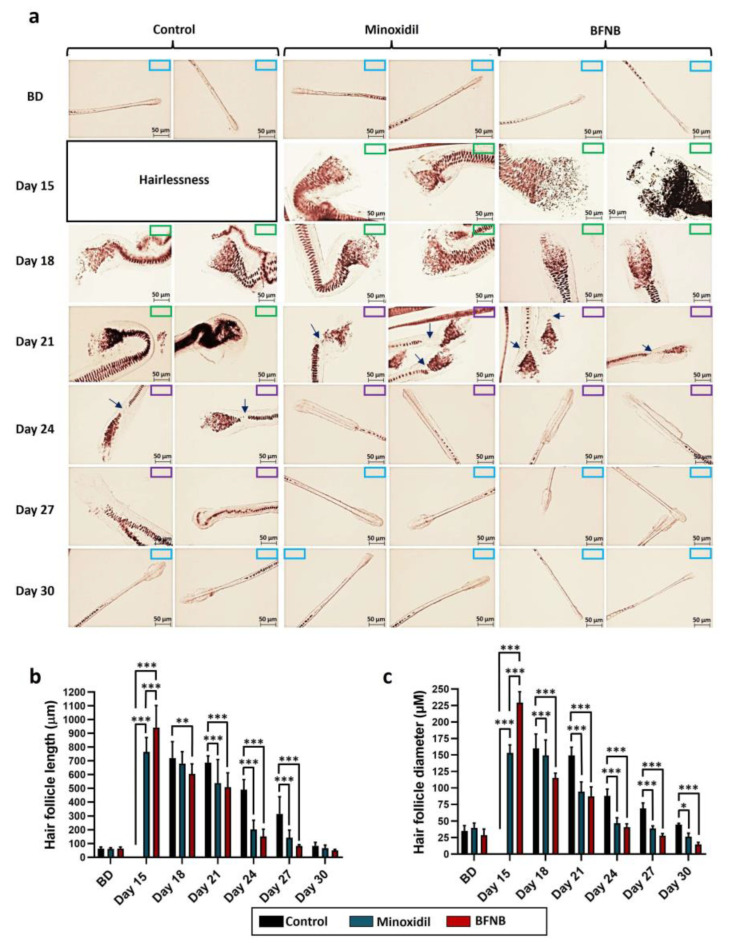
Effect of BFNB on the dynamics of follicular cycle phases in dorsal hairs. Hair morphology and progression of the hair cycle were assessed at 15, 18, 21, 27, and 30 days after treatment (**a**). The length (**b**) and diameter (**c**) of the hair follicles were measured on the indicated days. The initial hair length before treatment is denoted as BD. A color-coded system is used in the upper right corner of the images to identify the phases of the follicular cycle, with blue boxes representing the telogen phase, purple boxes indicating the catagen phase, and green boxes denoting the anagen phase. Arrows show the rupture of the inner root sheath cells, indicating the catagen phase. Scale bar = 50 µm. Statistical analysis of length and diameter was conducted using five hair follicles from each of the ten mice in each experimental group. One-way ANOVA was applied, followed by Tukey’s multiple comparison test. Comparisons between experimental groups are denoted by (*). The significance levels used were * *p* ≤ 0.033, ** *p* ≤ 0.002, and *** *p* ≤ 0.001.

**Figure 3 ijms-24-12110-f003:**
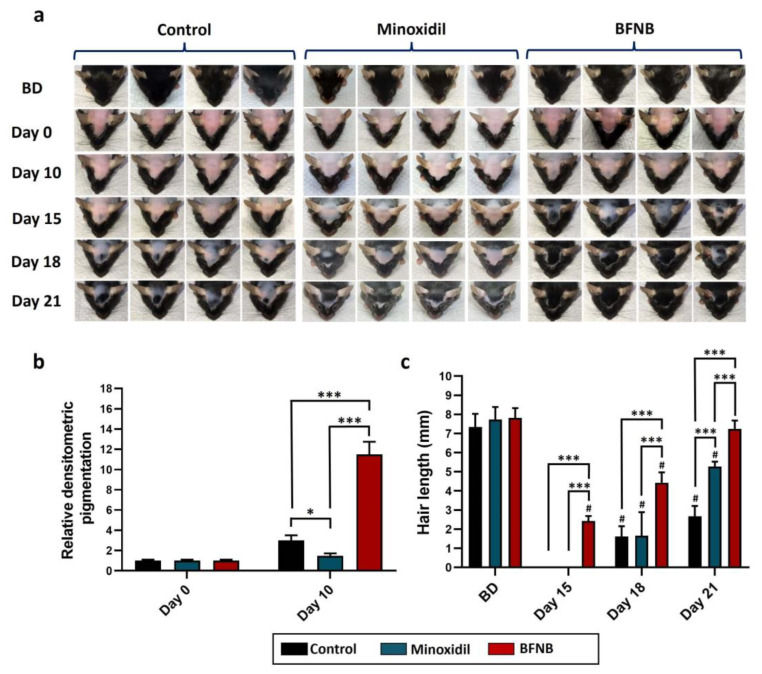
Promoter effect of BFNB on head hair growth in C57BL/6 mice. Evolution of hair growth during the 21 days of treatment (**a**). Densitometric analysis of head pigmentation was conducted on days 0 and 10 post-treatment (**b**). Measurements of hair length were performed 15, 18, and 21 days after treatment (**c**). The initial hair length before treatment is denoted as BD. Statistical analysis was carried out on the pixel-wise pigmentation data obtained from the skin images of 10 mice per experimental group. To determine hair length, five hair tufts were collected from each of the 10 mice in the different experimental groups. In both cases, a one-way ANOVA was applied, followed by Tukey’s multiple comparison test. The comparison between experimental groups is indicated by (*) and comparisons between groups on days 15, 18, and 21 with BD are denoted by (#). The significance levels are represented as follows: * *p* ≤ 0.033, *** *p* ≤ 0.001, and # *p* ≤ 0.033.

**Figure 4 ijms-24-12110-f004:**
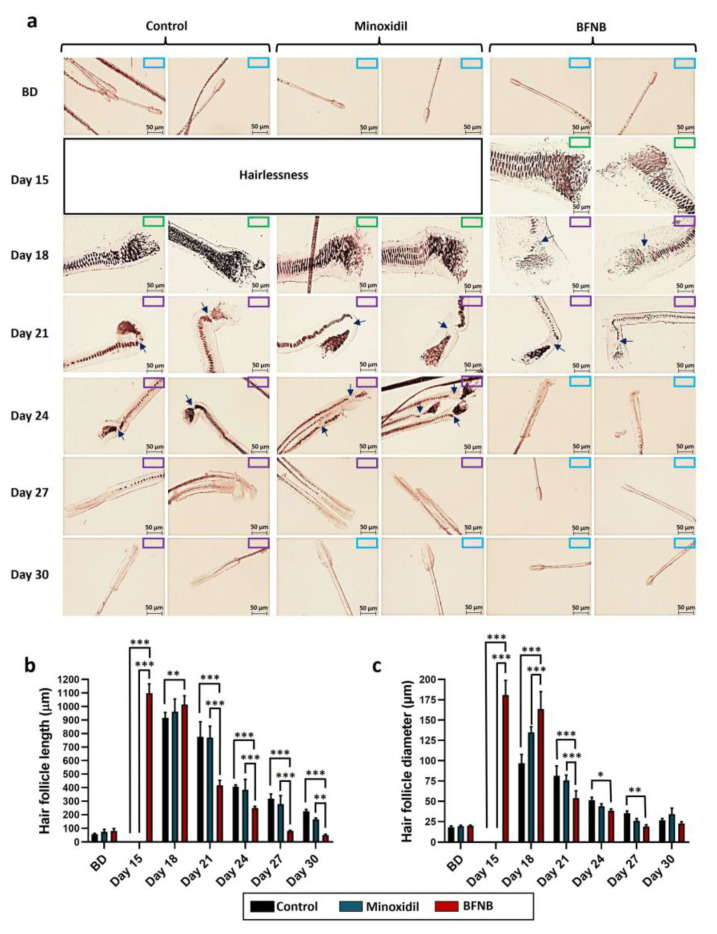
Effect of BFNB on the dynamics of follicular cycle phases in head hairs. Hair morphology and progression of the hair cycle were assessed at 15, 18, 21, 27, and 30 days after treatment (**a**). The length (**b**) and diameter (**c**) of the hair follicles were measured on the indicated days. The initial hair length before treatment is denoted as BD. A color-coded system is used in the upper right corner of the images to identify the phases of the follicular cycle, with blue boxes representing the telogen phase, purple boxes indicating the catagen phase, and green boxes denoting the anagen phase. Arrows show the rupture of the inner root sheath cells, indicating the catagen phase. Scale bar = 50 µm. Statistical analysis of length and diameter was conducted using five hair follicles from each of the ten mice in each experimental group. One-way ANOVA was applied, followed by Tukey’s multiple comparison test. Comparisons between experimental groups are denoted by (*). The significance levels used were * *p* ≤ 0.033, ** *p* ≤ 0.002, and *** *p* ≤ 0.001.

**Figure 5 ijms-24-12110-f005:**
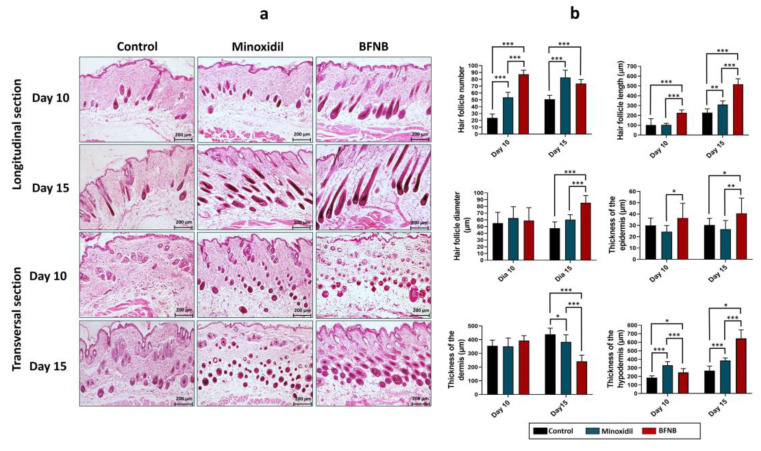
The architecture of the skin and dorsal hair follicles in C57BL/6 mice. Histological evaluation was performed by H & E staining on 8 µm thick longitudinal and transverse sections of mice skin, 10 and 15 days after treatment (**a**). The analysis consisted of quantifying the number, length, and diameter of hair follicles, as well as measuring the thickness of the epidermis, dermis, and hypodermis (**b**). Scale bar = 200 µm. Statistical analysis was performed by examining three fields per tissue, obtained from three mice of each experimental group. One-way ANOVA was applied, followed by Tukey’s multiple comparison test. Significant differences between experimental groups are indicated by the symbol (*). The significance levels used were * *p* ≤ 0.033, ** *p* ≤ 0.002, and *** *p* ≤ 0.001.

**Figure 6 ijms-24-12110-f006:**
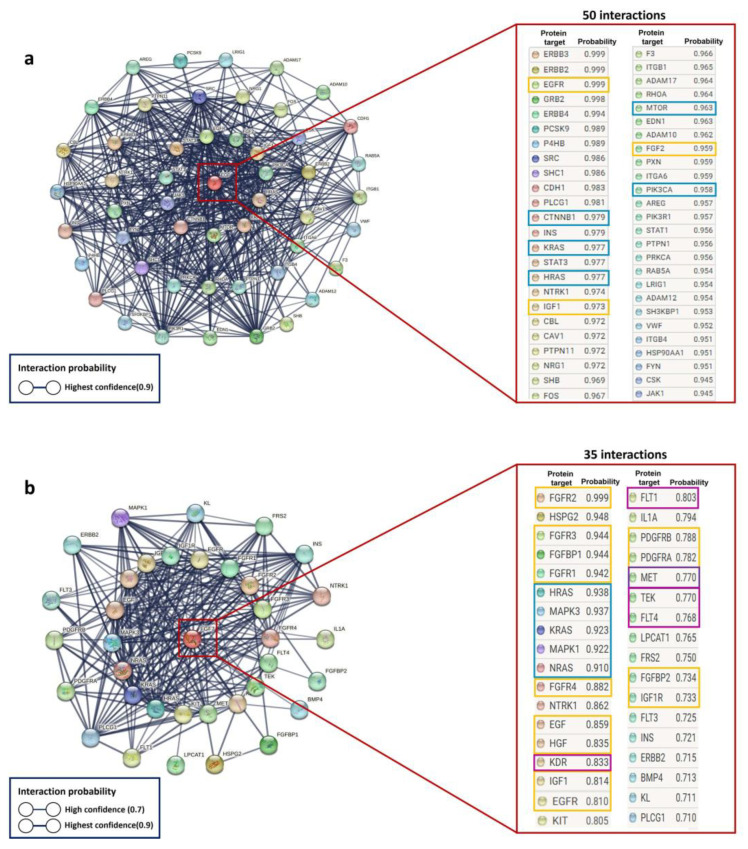
EGF and FGF growth factor interaction networks to predict key proteins and possible molecular pathways involved in hair growth. Interactomes of EGF (**a**) and FGF (**b**). The images illustrate a red box pointing to EGF and FGF proteins. The yellow boxes highlight growth factors, or their receptors related to hair growth, while the blue boxes represent proteins associated with the cell proliferation signaling pathway, and those in magenta play a role in modulating angiogenesis.

**Figure 7 ijms-24-12110-f007:**
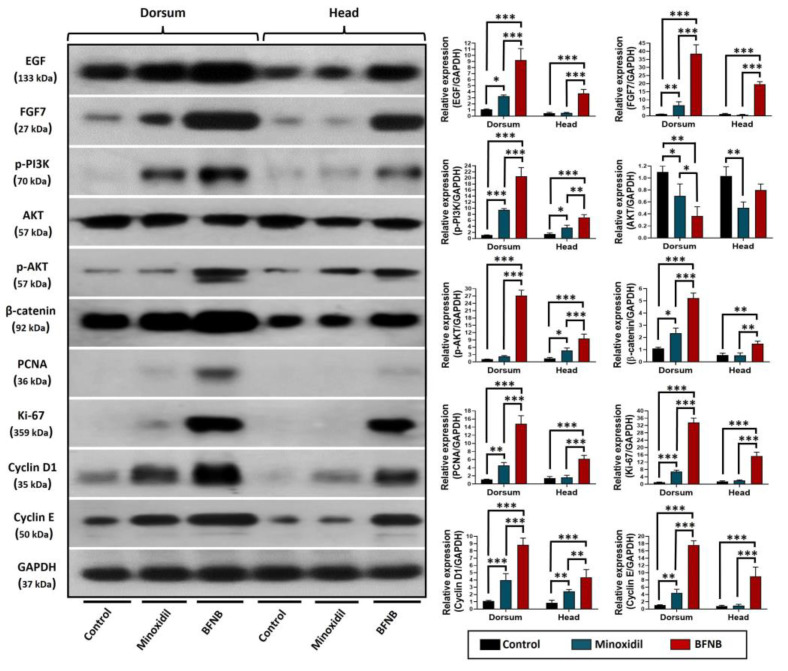
Effect of BFNB on the expression of growth factors EGF and FGF7, and proteins implicated in the PI3K-AKT pathway and cell cycle. Representative Western blots using protein extracts from the dorsal and head skin of mice after 15 days of treatment are shown in the left panel. The graphs show the relative expression of each protein evaluated. Expression was normalized using the expression of the endogenous control, GAPDH. Statistical analysis was performed using the average protein expression of three mice per experimental group. One-way ANOVA was applied, followed by Tukey’s multiple comparison test. Significant differences between experimental groups are indicated by (*). The significance levels used were * *p* ≤ 0.033, ** *p* ≤ 0.002, and *** *p* ≤ 0.001.

**Figure 8 ijms-24-12110-f008:**
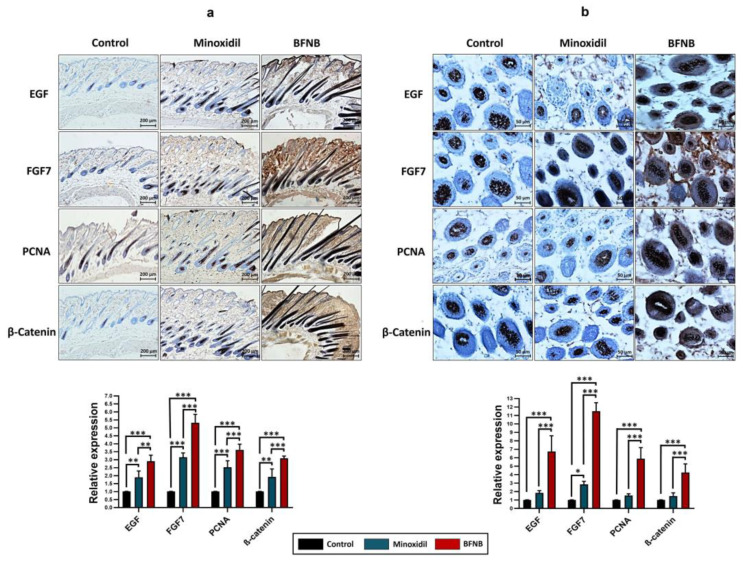
Immunodetection of the EGF, FGF7, PCNA, and β-catenin proteins in histological sections. Immunohistochemistry assays were performed on longitudinal (**a**) and transverse (**b**) dorsal sections of mice 15 days after treatment. Scale bar = 200 µm (**a**) and 50 µm (**b**), respectively. Graphs show relative protein expression, calculated as the average of pixel values obtained from three independent samples per experimental group. Statistical analysis was performed using Tukey’s multiple comparison test. Comparisons between experimental groups are indicated by (*). The significance levels used were * *p* ≤ 0.033, ** *p* ≤ 0.002, and *** *p* ≤ 0.001.

**Figure 9 ijms-24-12110-f009:**
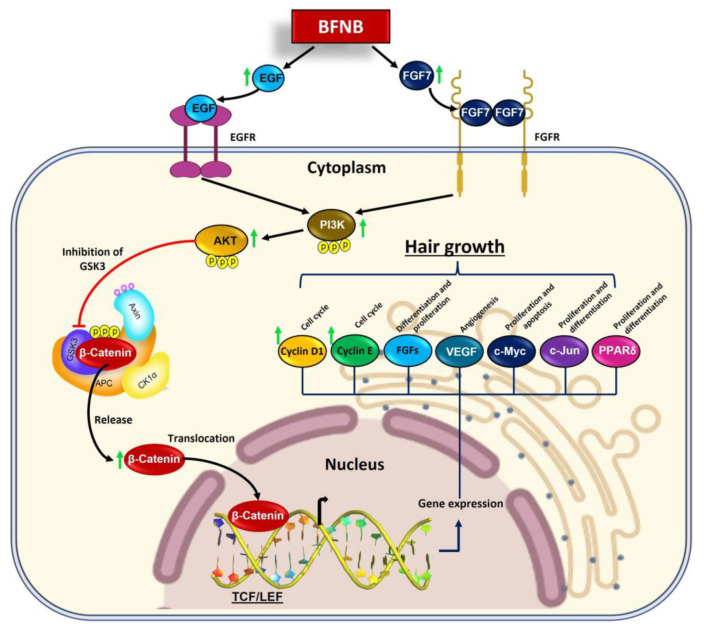
Possible molecular mechanisms modulated by BFNB metabolites to promote hair growth. BFNB induces the expression of EGF and FGF7 growth factors, probably in fibroblasts and keratinocytes of the dermis, as well as in hair follicle cells. Once synthesized and released by these cells, these factors lead to the activation of their receptors EGFR and FGFR7 in hair follicle cells. This results in the phosphorylation and activation of PI3K kinase. In turn, this protein promotes the activation of AKT through its phosphorylation. AKT blocks GSK3 kinase activity, allowing the release of β-catenin from its destruction complex. β-catenin is translocated to the nucleus to modulate the transcription of genes related to cell cycle, proliferation, cell growth, and differentiation, as well as angiogenesis, enhancing hair growth. Green arrows indicate key proteins whose expression increased in the dorsal and head regions after 15 days of BFNB treatment in C57BL/6 mice.

**Table 1 ijms-24-12110-t001:** Fold higher expression of BFNB compared to control or minoxidil (WB).

	Dorsum		Head	
	Control	Minoxidil	Control	Minoxidil
EGF	8.1	6	3.3	3.2
FGF7	37.4	31.9	18.5	18.9
p-PI3K	19.4	11.1	5.5	3.4
AKT	−0.7	−0.3	−0.2	0.3
p-AKT	26.1	24.9	8.2	4.9
β-catenin	4.1	2.9	0.9	1
PCNA	13.7	10.3	4.7	4.6
KI-67	32.6	26.8	13.9	13.3
Cyclin D1	7.7	4.9	3.5	1.9
Cyclin E	16.5	13.2	8.2	8.1

WB; Western blot.

**Table 2 ijms-24-12110-t002:** Fold higher expression of BFNB compared to control or minoxidil (IHQ).

	Dorsum		Head	
	Control	Minoxidil	Control	Minoxidil
EGF	1.9	1.0	5.7	4.9
FGF7	4.3	2.2	10.5	8.7
PCNA	2.6	1.1	4.9	4.4
β-catenin	2.1	1.2	3.3	2.8

IHQ; immunohistochemistry.

## Data Availability

The datasets used and/or analyzed during the current study are available from the corresponding author upon reasonable request.

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
