# Peer review of "BFNB Enhances Hair Growth in C57BL/6 Mice through the Induction of EGF and FGF7 Factors and the PI3K-AKT-β-Catenin Pathway"

_ijms, 2023, doi:10.3390/ijms241512110_

Round 1

Reviewer 1 Report

Dear authors,

Pérez-Mora et al. administrated BFNB extracted from Bacopa procumbens to mice and revealed that it has positively effects on hair growth with increase thickness of epidermis and hypodermis and acerating transition from telogen to anagen. In addition, the authors suggested that its hair growth was associated with upregulation FGF7 and EGF, and activation of their related pathways. These results suggested that BNFB is a useful compound for alopecia therapy. I felt results were supported by excellent experiments and data were clearly reflected authors’ claims. Thus, I recommend publishing this paper submitted by Pérez-Mora et al. in International Journal of Molecular Sciences. However, I think some minor points should be considered before publication.

Discussion

Point 1: L249, 268, 269, 276, 282, 305, 314, and 377.

Scientific names of these plants should be written by italic characters.

Point 2: L250.

The authors demonstrated that BFNB has the effect for hair growth. However, the author did not provide data of BFNB effects for hair health. If it is possible, please add description the reasons of the good effect for hair health. What is hair health?

Point 3: L281-282.

Please indicate how number of differences in follicle there were between R. officinalis and BFNB treatment.

Point 4: Materials and Methods, L424-425.

How did the author determine administration amount of BFNB? Have BFNB the effect dose-dependently in hair growth? If the authors have any data or speculation regarding dose-dependency of BFNB in hair growth, please provide information.

Point 5: Materials and Methods,

Please provide information about method of anesthesia during depilation and in hair observations.

Point 6: Figure 8, L240.

Scale bar= 200 µm (a) and 50 µm (b), respectively.

Reviewer 2 Report

The paper entitled "BFNB Enhance Hair Growth in C57BL/6 Mice through the In- 2 duction of EGF and FGF7 Factors and the PI3K-AKT-β-Catenin 3 Pathway" shows in a very clear manner how the topical application BFNB extract is able to increase in treated mice follicular cycle phases, the number of follicles, hair length, and diameter, compared to control mice or treated only with minoxidil. The manuscript presents very accurate techniques such as WB, IF, protein-protein interaction network which increase the scientific soundness of the results. Authors investigate many proteins and growth factors involved in epidermis, dermis and hypodermis differentiation. 

Suggestions: 

- it would be interesting for the authors to present some results also on  dermis capillarization, which is essential for the maintenance of the dermis-epidermis barrier. In fact the development of a vascular network is essential for the regulation of oxygen and nutrients in a tissue. Regarding the innervations or the other dermal appendages?

- could author explain better the functionalization of the gold nanoparticles generated to applied then subsequently the lipophilic serum. 

Minor editing of English language required. 

Reviewer 3 Report

The manuscript was aimed to demonstrate the effects of the bioactive fraction of nanostructured Bacopa procumbens (BFNB) on the hair growth. 

The authors demonstrated that BFNB was more effective compound to enhance hair growth when compared with minoxidil, a well-known drug used for the treatment of apolecia. The remarkable improvement in hair growth was observed in BNFB-treated mice, which was associated with acceleration of the transition process between the follicular cycle phases, and the increase of follicular sizes (length and diameter). 

The authors also demonstrated that BFNB's effects indicated above might be due to its ability to activate PI3K-AKT-catenin signaling pathway.  This was evidenced by increased levels of phosphorylated AKT and increased expression of beta-catenin. The authors proposed that the outcome of this activation would be the cell cycle dysregulation evidenced by the increased levels of the cyclines D1 and E and proliferative markers Ki-67 and PCNA.

In general, the manuscript is well-prepared. 

I have the following suggestions regarding this manuscript

1) The authors have to explain a minor (or even lack of) effect of minoxidil on the proliferative rate (PNCA and Ki-67 markers), as shown in Figure 7. 

2) Since an increased expression of FGF-7 was detected in BFNB-treated samples (Figure 7), it is recommended to include  the expression of the phosphorylated forms of FGFR1/2 and FRS-2 to define the specific molecular mechanisms of activation of AKT-mediated pathway. Similarly, expression of phosphorylated and total forms of EGFR is missing. 

3) Figure 9 requires an explanation and additional experiments. It is a well-known fact that FGFR family is composed of 4 types of receptors (FGFR1-4). In this case FGFR7 shown there looks as a typo. Moreover, Figure 7 illustrates the ability of FGF7 to bind with EGFR and further activate AKT-signaling pathway. It's a novel and too preliminary. The authors have to run WB for the activation markers of FGFR- and EGFR-signalling pathways, as was recommended above.  

4) Hyperexpression of FGF7 observed in BFNB-treated samples might be not related with its positive effect in vivo and this effect might be solely from the activation of EGFR-mediated pathway due to the  hyperproduction of the EGF ligand. The knock-down EGFR/FGFRs is required to deliniate between these possibilities.

5) Expression of the down-stream markers (total and phosphorylated forms of STAT, GSK and S6 protein) is also missing. 

Overall, despite the manuscript clearly illustrates the positive effect of BFNB in vivo, molecular mode of action of this compound is needed to be further examined in more detail. 

Round 2

Reviewer 3 Report

The authors responded to the comments and suggestions. The quality of the manuscript was improved.